# Factors Associated with Supportive Care Needs Among Palestinian Women with Breast Cancer in the West Bank: A Cross-Sectional Study

**DOI:** 10.3390/cancers16213663

**Published:** 2024-10-30

**Authors:** Ibtisam Titi, Nuha El Sharif

**Affiliations:** 1School of Public Health, Al-Quds University, Jerusalem 51000, Palestine; 2Ministry of Health, Ramallah P606, Palestine

**Keywords:** supportive care, breast cancer, services, unmet needs, Palestine

## Abstract

This study assessed the supportive care services offered to breast cancer (BC) patients in Palestinian government hospitals and identified characteristics related to unmet needs. We performed a cross-sectional study involving 362 women, utilizing the Supportive Care Needs Survey (SCNS-SF34) through face-to-face questionnaires, and gathered data on sociodemographic, clinical, and social support variables. The findings indicated that 61% of participants encountered unmet supportive care needs, especially in the areas of health system and information (93.6%), physical and daily living (89.8%), and psychological support (75.1%). Elevated unmet needs are associated with variables like age, marital status, illness duration, analgesic usage, cancer treatments, family cancer history, and social support. This research highlights the urgent need to enhance supportive care services of the health system and information, physical and daily living, and psychological support, to elevate the quality of life for BC patients in Palestine.

## 1. Introduction

Breast cancer (BC) is a prominent type of cancer that affects women globally. Regardless of the economic condition of a country, BC presents a significant challenge in terms of its prevention, detection, and treatment [1]. In 2022, there were an estimated 2.3 million newly diagnosed cases with BC and 660,000 related deaths globally [2]. In Palestine, BC is the most common and widely spread type of cancer. According to the Palestinian Ministry of Health’s annual report (2023), BC has an incidence rate of 18.5 cases per 100,000 of the total population and 37.4 cases per 100,000 of the female population, with a mortality rate of 11.7 per 100,000 of all cancers [3]. In the West Bank, 540 new BC cases were documented in 2022, accounting for 15.8% of all new cancer cases [3]. The BC mortality rate was higher in the southern governorates of the West Bank [4], with 32% of registered cases reported in Hebron and 13% in the Bethlehem governorate [5].

The term “supportive care needs” is an umbrella word that covers the physical, informational, emotional, practical, social, and spiritual needs of a person with cancer [6]. It is a patient-centered approach aimed at supporting cancer patients and their families by addressing their needs at various stages, including pre-diagnosis, throughout diagnosis and treatment, during remission, in chronic disease, end-of-life care, and the grieving process [7]. Additionally, supportive care focuses on optimizing treatment benefits to improve patient survival rates and quality of life. Supportive care in cancer treatment offers significant benefits, such as reduced morbidity, and it can benefit healthcare services by reducing the use of healthcare resources and improving overall treatment outcomes [7,8,9].

There is a growing interest in conducting supportive care needs assessments to ensure that patient needs are met. “Unmet needs” are defined as needs that are not sufficiently met and need more support [6]. Several studies have identified key needs commonly experienced by BC patients, which encompass various areas such as treatment information, symptom management, medical guidance or counseling, psychological support, effective communication with healthcare systems, daily care assistance, support for reintegrating into society, access to resources, substance support, and emotional and spiritual well-being [9,10,11,12,13]. In addition to recognizing BC patients’ supportive care needs, it is critical to identify the patient’s main characteristics to determine the unmet needs predictors. Several studies connect unmet supportive care needs with patients who have psychological concerns, including depression, anxiety, distress, and fears about cancer spread [11,14,15,16].

Several studies have shown a significant association between higher levels of unmet needs and factors such as younger age, higher education, unemployment, shorter survivorship duration, undergoing active treatment, advanced disease stage, and shorter time since diagnosis [17,18,19]. However, in Palestine, very few studies have focused on the supportive care needs of BC patients. One mixed-method study on BC patients in the Gaza Strip revealed significant gaps in meeting care needs, notably in psychological support, health system information, and physical care [19]. However, no studies in the West Bank have evaluated the supportive care needs of these patients. This information gap must be addressed, as understanding the unique challenges and requirements of patients from the time of diagnosis is key to improving their care and overall well-being. Therefore, this study aims to assess the supportive care services provided at the governmental hospitals in the southern area of the West Bank and to determine the factors associated with the unmet needs of these services.

## 2. Materials and Methods

### 2.1. Study Design

This is a cross-sectional study on supportive care needs among women with BC in the West Bank. In this design, data were collected from BC women receiving care in governmental hospitals in the southern area of the West Bank. This research design is suitable for performing population-based studies to estimate prevalence and identify risk variables. Moreover, it is advantageous for planning, monitoring, and evaluating public health measures.

### 2.2. Participants and Setting

The total population of the West Bank is 3,291,406. The southern area of the West Bank encompasses two out of eleven governorates: Bethlehem (total population: 247,197) and Hebron (total population: 832,702). The Bethlehem governorate has a total population of 247,197 and the Hebron governorate has a total population of 832,702. This area comprises around 40% of the West Bank population of whom 50% are women [20]. The Ministry of Health (MoH) is the primary healthcare provider for cancer patients.

The MoH collaborates with the United Nations Relief and Works Agency (UNRWA), Non-Governmental Healthcare Organizations (NGOs), and the private sector to promote cancer prevention and control [5]. All cancer cases in the southern area of the West Bank are referred to two governmental hospitals located in the Bethlehem governorate and Hebron governorate. The three hospitals provide diagnosis, treatment, and inpatient services.

### 2.3. Sampling Method and Sample Size

An online sample size calculator (https://www.calculator.net/sample-size, accessed on 22 October 2024) estimated the required sample size to be 333, using a 5% error and a power of 95% confidence interval, assuming that 50% of females with BC had unmet supportive care needs. We added a 10% non-response rate to the 333 patients, resulting in a final sample size of 362 women. A complete list of all registered BC patients from the three hospitals was obtained. Eligibility for this study was restricted to newly registered BC cases and those currently undergoing medical treatment. The required data were retrieved from the patients’ medical records. Women were contacted during their weekly or monthly hospital visits and invited to participate. Women who accepted to participate were interviewed on the day of their hospital visit. Only two women refused to participate.

### 2.4. Data Collection

To assess the predictors for factors affecting patients’ unmet needs, we developed an additional questionnaire to collect patients’ demographic data, disease-related information, socioeconomic status, age, cultural norms, educational background, beliefs, personality, cancer stage, residence area, and levels of social and familial support. The questionnaire was filled out through face-to-face interviews. Data were collected between March and July 2024. The questionnaire was created using the Kobo Toolbox software (https://www.kobotoolbox.org), a free toolkit for collecting, managing, and visualizing data in challenging environments [21]. We also utilized the available data from patient medical files.

Supportive care needs were measured using the short form of the Supportive Care Needs Survey (SCNS-SF34). The SCNS-SF34 is a widely used instrument designed to assess the supportive care needs of cancer patients. It is a shorter version of the original Supportive Care Needs Survey. The SCNS-SF34 contains 34 items covering five domains of need: psychological, health system and information, physical and daily living, patient care and support, and sexuality. Each item is scored on a 5-point Likert scale, ranging from 1 (“not applicable”, “no need for help”) to 5 (“very high need for help”), with higher scores indicating greater unmet needs. The SCNS-SF34 is scored using a Likert scale for each item, ranging from 1 (no need) to 5 (high need). To calculate standardized domain scores, the sum of scores for items within each domain is computed, subtracted by the number of items in the domain, and then multiplied by 100 over the product of the number of items and the maximum response value minus one. This scoring system provides a range of possible values from 0 to 100 for each domain, with higher scores indicating greater unmet needs [22].

We used the Arabic version of the Supportive Care Needs Survey (SCNS-SF34). The supportive care needs questionnaire was translated into Arabic and then back-translated into the English language. Three healthcare services and cancer care experts validated the study tool for its content validity. For reliability, we carried out a pilot study on 20 patients. The overall Cronbach’s alpha of the SCNC-34 for the five domains was 0.957 (ranging from 0.860 to 0.961: physical and daily living needs domain = 0.909, psychological needs domain = 0.961, sexual needs domain = 0.860, patient care and support needs domain = 0.917, and health system and information needs domain = 0.958).

### 2.5. Statistical Analysis

Data analysis was conducted using the Statistical Package for the Social Sciences (SPSS), version 25. We utilized descriptive statistics to summarize the sociodemographic and clinical characteristics of the participants. Continuous variables, including age, were categorized into three groups, while the number of children and family size were categorized into two groups. For statistical analysis, marital status was divided into three categories, education into four categories, work status, and diagnosis duration into three categories (see Table 1 and Table 2). A Chi-square test was used to investigate the associations between the domains of the Supportive Care Needs Survey-Short Form 34 (SCNS-SF34) and the sociodemographic and clinical characteristics of the participants. A *p*-value of less than 0.05 was considered statistically significant.

We reclassified the five supportive care domains according to McElduff and colleagues, who presented two scoring systems. The first one depended on summing individual items within each domain (summated scale), while the second created standardized total scores for each domain, with scores ranging up to 100, where a higher score indicated a greater perceived need. In this study, we adapted the first scoring system [22]. Consequently, we converted the patients’ responses on the 5-point SCNS-34 questionnaire to a 2-point scale, categorizing them as either “No needs” or “Unmet needs”. “Unmet needs” included the low, moderate, or high needs, and those with “No needs” included those with no need or were satisfied with supportive care services. Therefore, we classified patients with scores of three or higher as having unmet needs, while those with scores of one or two had no need.

The prevalence of supportive care needs was calculated for each domain and the overall scale. Responses of participants to the SCNS were summed for each domain to determine the domains and the overall prevalence. Coding was also applied as 1 “no needs” and 2 “unmet needs”. We conducted a logistic regression analysis to examine the associations between the independent variables (sociodemographic and clinical characteristics) and the dependent variables (domains of unmet supportive care needs). The adjusted odds ratio (AOR) and its 95% confidence interval are reported.

### 2.6. Ethical Considerations

We obtained approval from the Al-Quds University Research Ethical Committee (reference number REF. 10/24) and approval from the Health Education and Scientific Research Department of the Ministry of Health (reference number 162/475/2024). In addition, all participants were informed about the study objectives and signed an informed consent form emphasizing participant confidentiality and their right to withdraw from the study at any time without impacting their clinical treatments.

## 3. Results

### 3.1. Sociodemographic and Clinical Characteristics of the Participants

Table 1 displays participants’ demographic characteristics. This study included a cohort of 362 female patients diagnosed with BC. Participants’ mean age was 48.83 (SD ± 10.13) years and 51.4% were in the age group 41–54. The majority (84.3%) were married, and 67.4% were from the Hebron governorate. Furthermore, 67% of the participants had five or fewer children, and 63.5% lived in families with more than five members. Additionally, 45% of the participants completed their secondary education, 29.3% had a university education and above, 77.3% were housewives, and only 20.2% were employees. The participants’ family income varied, with 45.0% receiving a monthly salary of less than 570 US dollars.

### 3.2. Clinical Characteristics of the Participants

Table 2 presents the participants’ clinical characteristics. Of patients, 61.6% were diagnosed more than a year ago, 47.5% were diagnosed at stage 3, 40.2% underwent a total mastectomy, and 34.8% reported having chronic diseases other than cancer (thyroid disease, diabetes mellitus, heart disease, and hypertension). Approximately 50% of the patients indicated the use of painkillers and had a familial history of cancer, with 27.6% specifically having BC. Out of all the participants, only 2.8% (10 people) were referred to care providers who were not cancer specialists. These care providers included eight dietitians, one physiotherapist, and one ophthalmologist. Concerning the support patients received during their illness journey, the majority of them (84.5%) indicated receiving support from their family and sons/daughters, while 25.7% received support from their husband or partner, and only six individuals (1.7%) reported receiving support from medical staff (see Table 2).

### 3.3. Supportive Care Needs Using the SCNS-34 Short Form

The results revealed that the participants expressed the highest need for support for the health system and information domain (mean 39.64, SD 10). However, they had the least need for support in the sexual need domain (mean 6.31, SD 3.36) (see Table 3).

#### 3.3.1. Physical and Daily Living Needs

Based on the classified scale, the overall mean score for the physical and daily living domains was 17.98 (SD ± 5.00). The vast majority of participants needed help with work around the home (89.0%), 86.5% felt unwell a lot of the time, 85.9% were unable to do things they used to do, 78.2% complained of a constant feeling of lack of energy/tiredness, and 59.9% were in need of support related to pain management (see Figure 1).

#### 3.3.2. Psychological Needs

Furthermore, the overall mean score for the psychological need domain was 29.22 (SD ± 10.48). Of the participants, 64.4% reported experiencing anxiety, 63% expressed fears about the cancer spreading, and 62.7% reported feeling down or depressed. Furthermore, 61.6% expressed worries that the treatment results were beyond their control, 61.3% expressed sadness, and 60.5% of participants expressed uncertainty about the future (see Figure 2).

#### 3.3.3. Sexual Need

In addition, the overall mean of the sexual need domain score was 6.31 (SD ± 3.36). Of the participants, 68.5% felt there was no need to be given information about sexual relationships, 36.5%, expressed a need for information about sexual relationships, and 34.5% reported experiencing changes in their sexual feelings (see Figure 3).

#### 3.3.4. Patient Support and Care Needs

Moreover, the overall mean score for the patient support and care needs domain was 16.34 (SD ± 5.16). The most notable need identified of participants was for hospital staff to acknowledge and demonstrate sensitivity to patients’ feelings and emotions (80.1%). Hospital staff’s prompt attention to participants’ physical needs ranked second (78.2%), 74% needed reassurance from medical staff that what they felt was normal, 61.9% of participants required more choices regarding the hospital they attend, and 57.5% required more choices about their cancer specialists (see Figure 4).

#### 3.3.5. Health System and Information

Finally, the overall mean score for the health system and information needs domain was 39.64 (SD ± 10.0). Of the participants, 87% expressed a desire for more information and an explanation of the tests they must complete. Additionally, 86.70% of the participants expressed a need for written information about managing their illness and side effects at home; 84.0% expressed a need for written information about important aspects of their care; and 79.30% expressed a need for information about whether their cancer is under control or diminishing. The lowest reported need in this domain was the desire to be treated like a person, not just another case, with 69.3% of participants expressing this concern (see Figure 5).

### 3.4. Prevalence of Supportive Care Needs (SCNS-34)

Figure 6 summarizes the overall prevalence of supportive care needs across all domains. The analysis reveals significant unmet needs in all domains. A total of 61% of respondents reported unmet needs, and 39% reported no need for overall supportive care. The health system and information domain had the highest unmet needs (93.6%), followed by physical and daily living needs (89.8%), patient care and support (81.80%), psychological needs (75.1%), and sexual needs (42.5%).

### 3.5. Bivariate Analysis

#### 3.5.1. Supportive Care Needs and Sociodemographic Factors

The study findings show that participant age was significantly associated with all supportive care needs domains (*p*-value < 0.05). However, the participant’s governorate, place of residence, and marital status did not show any significant association with supportive care needs domains (*p*-value > 0.05) except that patient care and support care needs domains were significantly associated with place of residence (*p*-value = 0.033), and sexual care needs domain was significantly associated with marital status (*p*-value = 0.001) (see Appendix A).

The participant’s level of education was significantly associated with all supportive care needs domains (*p*-value < 0.05), except for the physical and daily living needs (*p*-value = 0.72). Results showed a significant association between patients’ educational level with psychological needs domain (*p*-value < 0.001), sexual care needs domain (*p*-value < 0.001), patient care and support needs domain (*p*-value = 0.004), and health information needs domain (*p*-value = 0.04), with unmet needs being more common among those with secondary education. The findings also demonstrate that working status was only significantly associated with psychological needs care (*p*-value = 0.004) and sexual care needs (*p*-value = 0.01) compared to those working or retired women. Moreover, patients who are housewives had higher percentages of unmet needs in all supportive care needs domains compared to those working or retired women (see Appendix A).

Furthermore, results showed that families with more than five members are more likely to have unmet physical care needs (65.2%) (*p*-value = 0.04) (see Appendix A), and women with a family monthly salary below USD 570 are more likely to have unmet sexual needs (*p*-value = 0.03) and to have unmet patient care and support needs (*p*-value = 0.014) compared to those with higher incomes (see Appendix A).

#### 3.5.2. Supportive Care Needs, Clinical, and Social Support Factors

The study results showed a significant association between BC duration and the psychological and sexual care needs domains (*p*-value < 0.001); patients with BC for more than a year had higher unmet needs compared to others. Also, findings revealed that individuals using pain medication had more unmet physical care needs (*p*-value < 0.001) (see Appendix A). Furthermore, 36.6% of patients with chronic diseases had unmet physical and daily living needs, 29.8% psychological care needs, 26.0% sexual care needs, and 32.4% patient care and support care needs (*p*-value < 0.05) (see Appendix A).

Cancer therapy was significantly associated with supportive care needs. Results also revealed that chemotherapy was associated with higher unmet physical and daily living care needs (*p*-value = 0.03). Moreover, radiation therapy is associated with higher psychological (*p* = 0.02) and sexual unmet care needs (*p*-value = 0.001) (see Appendix A). Also, patients not receiving hormone therapy had higher unmet psychological and sexual needs (*p*-value < 0.05) (see Appendix A). On the contrary, patients receiving hormone therapy had higher unmet patient care and support needs (*p*-value = 0.009) (see Appendix A). In addition, patients without biological therapy (immunotherapy) had higher unmet sexuality needs (*p*-value = 0.001), while those who underwent surgical therapy had higher unmet sexuality needs (*p*-value = 0.04). Finally, lack of support from a husband or partner is strongly associated with unmet sexual needs (*p*-value < 0.001) (see Appendix A).

### 3.6. Multivariate Analysis

Table 4 presents the multivariate analysis for the determinants of supportive care needs domains and the total supportive needs domains models. The results showed a strong association between age and all supportive care needs domains. Younger age groups (≤40 years and 41–54 years) had lower odds of physical and daily living care needs (AOR = 0.21, *p*-value = 0.01; AOR = 0.31, *p*-value = 0.04, respectively) compared to older age groups (≥55 years). On the other hand, younger age groups had significantly 13 times and 5.84 times higher odds, respectively, for the unmet psychological care needs compared to older age groups. Furthermore, these groups were also associated with higher odds of unmet sexual needs, patient care and support care needs, and health information unmet care needs compared to older ages. Also, married participants and individuals with a familial history of BC had higher sexual needs for care (AOR = 4.62, *p*-value = 0.01) and (AOR = 2.34, *p*-value = 0.006) in comparison to divorced or widowed women. The probability of needing physical and daily living care demands was 4.5 times greater with chemotherapy (AOR = 4.49, *p*-value = 0.03) than without it (Table 4).

Moreover, the findings indicated that a BC diagnosis for less than 6 months (AOR = 3.63, *p*-value = 0.04) and between 6 and 12 months (AOR = 8.49, *p*-value = 0.045) is associated with an increased likelihood of needing physical and daily living care support. Furthermore, the use of painkillers doubles the likelihood of physical and daily living needs by almost five times and doubles the overall care demands. In addition, surgical intervention increases the odds of physical and daily living needs by over fivefold (OR = 5.15, *p*-value = 0.00), whereas partial mastectomy increases the likelihood of overall care needs by more than twofold (AOR = 2.27, *p*-value = 0.01) in comparison to no surgical intervention. However, hormonal treatment reduces the odds of psychological needs (AOR = 0.36, *p*-value < 0.001) and the odds of patient care and support needs (AOR = 0.48, *p*-value = 0.01) compared to not having a hormonal treatment. Also, radiotherapy lowers the odds of sexual care needs by 58% (AOR = 0.42, *p*-value = 0.003), and biological therapy lowers it by approximately 60% (AOR = 0.40, *p*-value = 0.001) compared to not having radiation or biological therapy (Table 4).

Finally, family and sons/daughters support decreases the likelihood of physical and daily living needs (AOR = 0.19, *p*-value = 0.03) and total needs (AOR = 0.50, *p*-value = 0.05) compared to patients without family and son support. Interestingly, a family history of BC increased the likelihood of sexual care needs twofold compared to those without a family history of BC (Table 4).

## 4. Discussion

This study is the first to examine the prevalence and factors associated with unmet supportive care needs among breast cancer patients in the West Bank of Palestine. Our findings reveal a significant deficiency in the supportive care services at Palestinian Ministry of Health facilities, particularly with respect to information and psychological needs.

Overall, 61% of participants experienced unmet needs, with the most prevalent needs in the health system and information domain (93.6%), followed by physical and daily living needs (89.8%), and psychological needs (75.1%). These results align with previous research on supportive care challenges faced by breast cancer patients worldwide, especially in low- and middle-income countries (LMICs). In LMICs, cancer presents a major issue, and insufficient supportive care services can exacerbate patients’ stress and their inability to cope with the psychological consequences of diagnosis and treatment. For example, patients may struggle to access necessary medications, understand their treatment options, or find reliable sources of information [23,24]. Cultural perspectives can also influence the types of supportive care needs that patients prioritize. For instance, in Malaysia, a study found that fewer patients expressed fear of cancer spreading compared to patients in other regions. However, there was a higher demand for written information and additional explanations about treatment procedures [18].

In Palestine, the lack of integrated and multidisciplinary care teams at Ministry of Health facilities hinders the delivery of comprehensive supportive care. This can lead to fragmented care and delays in treatment. Furthermore, the lack of explicit procedures or guidelines for cancer-supportive care can intensify these difficulties. Consequently, numerous patients may not receive sufficient assistance to tackle the physical, mental, and social challenges associated with breast cancer. This may significantly impact their quality of life, resulting in heightened anxiety, stress, and depression. Patients may also encounter challenges in sustaining their daily habits, managing their finances, and preserving relationships with loved ones. Although the MoH is responsible for providing cancer care, the political conflict in the region has resulted in a deterioration in the Ministry of Health’s financing, putting the provision of essential health services for cancer patients at risk due to economic constraints. Therefore, collaboration with other healthcare providers or institutions can enhance the delivery of comprehensive services in regions where the Ministry of Health faces service shortages. Additionally, receiving support from donor organizations through financial and technical assistance will be advantageous.

### 4.1. Supportive Care Needs Sub-Domains

Our study reveals significant unmet needs across the five domains of supportive care, with the health system and information domain showing the highest unmet needs at 93.6%. These findings exceed those of two local studies conducted in the Gaza Strip to evaluate the supportive care needs of cancer patients [19,25]. Conversely, our findings are consistent with previous studies in Singapore [26], Germany [27], Mexico [15], and China [28], where the health system and information domain also accounted for the highest unmet needs. This suggests many patients may lack adequate knowledge about their disease and treatment options. Additionally, there is a significant need for written information on disease management (87%) and additional explanations for their laboratory tests (86.7%). Factors such as limited access to specialized healthcare providers, inadequate information about treatment options, and financial constraints may contribute to the high unmet needs in this domain. Although doctors and nurses strive to address patient’s needs, they often lack specialized training, resources, and enough time to provide the necessary support. Additionally, Palestine has unique socio-political and economic challenges that may impact the availability and accessibility of healthcare services, including those related to cancer [29,30].

Participants reported a high prevalence of unmet needs (89.8%) in the physical and daily living domain, including frequent feelings of unwellness, persistent fatigue, and difficulty with pain management. These findings align with two local studies conducted in the Gaza Strip, which reported similarly high rates of unmet physical needs (91%) and specific symptoms such as chronic pain, muscle weakness, and sleep disturbances [19,25]. Our results exceed those of an Iranian study, suggesting that the challenges faced by our participants may be particularly severe [17]. These physical needs represent challenges with pain management, fatigue, mobility, and assistance in tasks like housekeeping and childcare. Furthermore, we found a negative correlation between physical function and symptoms like pain and fatigue, consistent with previous studies on these issues [31]. The higher unmet needs observed in our study may be attributed to the older age of our patients or the side effects of the disease and its treatments.

In the psychological needs domain, 75.1% of patients reported unmet needs. More than two-thirds experienced anxiety, depression, and fear of cancer recurrence, highlighting the significant psychological distress faced by these individuals. While our results align with previous studies that have identified high levels of psychological distress among cancer patients, there are variations across different regions. For example, studies conducted in the Gaza Strip reported lower percentages of worries about treatment outcomes and fears of cancer spreading. However, Elsous and colleagues found that psychological needs were the most significant unmet needs in their study, with higher rates of anxiety (75%), sadness (73%), and worries about close relationships (87.8%) [19,25]. Globally, our results exceed those of many studies, particularly in terms of anxiety, depression, and fear of cancer spreading as seen in Switzerland [32] and in Turkey, where three out of five BC patients may experience depression and one out of every five patients may suffer from anxiety [33]. Similarly, Fan et al.’s analysis of 77 studies revealed lower levels of worry over treatment outcomes being beyond one’s control (54%), anxiety (48.7%), and fear of cancer recurrence (47.9%) [6]. However, a study in Malaysia reported even higher levels of psychological distress, including uncertainty about the future (78.6%), concerns about cancer spreading (76.1%), feelings of sadness (69.2%), and feelings about death and dying (68.4%), suggesting that the challenges faced by cancer patients can vary significantly across different cultural contexts [31].

Several factors may contribute to the high levels of unmet psychological needs, including fear of death, traumatic events, uncertainty about the future, family concerns, and financial stress. These challenges can make it difficult for patients to focus on recovery and maintain their mental well-being. Addressing these psychological needs is essential for improving the overall quality of life.

In the sexual care needs domain, 42% reported unmet sexual needs, with 34.5% experiencing changes in their sexual feelings such as decreased libido, pain during intercourse, or difficulty achieving orgasm. These findings align with a local study in the Gaza Strip, which identified sexual needs as a less frequently reported area of concern (15%) among breast cancer patients [19]. Several global studies have identified the sexuality domain as the least significant and least important among patients [31,34]. Furthermore, descriptive correlational research in Iran reported a similar percentage of unmet sexual care needs, indicating that this domain is more commonly identified as a concern in some regions [17]. In our study, sexual needs were the least unmet, possibly due to cultural norms, a lack of awareness, and the limited availability of cancer services. Marriage can increase concerns about sexuality-related issues, particularly in the Palestinian context, where cultural norms often cause females to feel uncomfortable or embarrassed when discussing their sexual needs. This may lead to underreporting of unmet sexual needs, as patients may be hesitant to disclose their concerns to healthcare providers. Addressing unmet sexual needs requires a sensitive and supportive approach that takes into account cultural factors and the specific needs of individual patients. Providing access to specialized healthcare professionals who can address sexual concerns and offer appropriate interventions can help improve the quality of life for breast cancer patients.

### 4.2. Sociodemographic and Supportive Care Domains

Sociodemographic characteristics had a significant role in predicting unmet needs for BC patients. This study revealed an association between age and all domains of supportive care needs. We found that patients who are 55 years old or older have significantly higher physical and daily living needs than younger patients. These results are in contrast with the Abdollahzadeh et al. study, which emphasized that younger BC patients have higher unmet needs in all supportive care domains [17]. Similarly, a Malaysian study reported that respondents under age 60 reported significantly higher unmet physical and daily living needs [18]. Older patients often have pre-existing health issues such as heart disease, diabetes, or arthritis, which can complicate BC treatment. Furthermore, as people age, their physical strength, endurance, and flexibility can change, making it more difficult to deal with the physical adverse effects of treatment. For example, older patients may experience fatigue, pain, and difficulty with mobility, which can impact their daily activities and overall quality of life.

Moreover, we found that younger patients and married patients reported higher unmet sexual needs compared to older and single participants. Our study findings are consistent with previous studies examining the impact of age and marital status on unmet sexual needs. For example, an Australian study of 444 adult cancer survivors found that younger patients had higher unmet sexual care needs, and married survivors were three times more likely to report unmet sexual needs than single or widowed survivors [35]. Similarly, in a Swiss study, married BC patients had higher unmet sexual needs than single individuals and that younger patients also reported higher unmet sexual needs [32]. Additionally, a broader review of unmet needs in cancer patients found that younger age and marriage are consistently related to larger unmet needs in the sexuality domain [36]. These findings can be explained by the fact that BC can disrupt these young patients’ life goals, leading to confusion and distress. They may also have concerns about hair loss and having a mastectomy, which can negatively impact self-esteem and body perception. Furthermore, worries about the effects of medication on fertility and reproduction may contribute to their unmet needs. Married patients prioritize sexual health due to active relationships, which may highlight intimacy challenges caused by cancer therapies. This raises the need for an awareness of unmet sexual needs.

Additionally, our study indicates that younger age groups revealed significantly higher levels of unmet psychological and sexual needs. These findings align with previous research, including a local study from Gaza, which found that patients under the age of 40 need more extensive care and support, particularly in psychological and sexual needs, compared to older patients [19]. Furthermore, Lidington et al. (2021) conducted a comprehensive longitudinal study in the United Kingdom, which found that younger individuals diagnosed with BC exhibited the highest level of need in the psychological and sexual domains [37]. Similarly, recent cross-sectional research in Turkey, which supports these findings, revealed that younger women with BC required more supportive care, particularly in the areas of psychological and sexuality [38]. Therefore, younger BC patients experience more psychological distress due to life milestone disruptions, body image and fertility concerns, and isolation from being diagnosed with a condition typically associated with older age groups.

### 4.3. Clinical Characteristics and Supportive Care Domains

Our findings revealed that BC patients with less than 6 months of diagnosis and those with 6–12 months of diagnosis had higher physical and daily living needs. This aligns with previous research demonstrating a strong association between time since diagnosis and unmet physical and daily living needs in BC patients. In Gaza, BC duration was negatively associated with unmet needs, and those just diagnosed had the largest unmet needs for up to three years following diagnosis [19]. Studies from various regions, including Malaysia, Taiwan, Denmark, and China, consistently report higher unmet needs in newly diagnosed patients. These needs often persist for several years post-diagnosis, particularly during the initial treatment phases [18,28,31,39,40]. In the Taiwanese retrospective study, cancer survivors’ supportive care needs decreased with time, with considerably lower needs through treatment follow-up compared to the newly diagnosed stage [39]. Additionally, in Denmark, BC patients receiving chemotherapy had higher unmet physical and daily living needs than those receiving radiation, especially in the initial treatment phase, indicating that unmet needs are higher within the first six months of diagnosis [40]. Similarly, in China, each 6-month increase in the duration since diagnosis led to a 0.8% decrease in the ongoing care needs [28]. These findings are consistent with a comprehensive review of 77 studies that revealed an association between short periods of diagnosis and increased unmet needs regarding both physical and psychological dimensions. [6]. Common challenges associated with early-stage breast cancer include fatigue, pain, and limited mobility, which can significantly impact daily activities. Various cancer treatments, such as surgery, chemotherapy, radiation, and hormone therapy, can exacerbate these issues and contribute to unmet needs. The cumulative effect of these factors often hinders patients’ ability to meet their physical and daily living needs [27,32,35,38].

Chemotherapy is a well-documented risk factor for unmet needs in breast cancer patients, particularly in the physical, daily living, psychological, and sexual domains [34]. Our study revealed that patients who received chemotherapy were 4.5 times more likely to experience an increase in unmet physical and daily living needs. In Gaza, chemotherapy treatment for BC and having at least one comorbid condition were associated with worse physical and social functioning, increased discomfort, poorer sleep, and fatigue [19]. Our study has similar findings to previous research from various regions, including Australia and Turkey, which consistently demonstrates that chemotherapy is associated with increased physical and daily living challenges [35,38]. Common side effects of chemotherapy, such as fatigue, nausea, and weakness, can significantly impair patients’ ability to perform daily activities and meet their needs. These challenges are often more pronounced in younger patients and those who receive both chemotherapy and radiation.

Additionally, the study multivariate analyses showed that chronic diseases increased the burden on BC patients, resulting in higher physical and daily living unmet needs and worsening the challenges associated with BC care. Several studies showed that comorbidity, or multiple medical conditions, is a common occurrence among cancer patients and is becoming more common with increasing age [41,42]. For instance, Fu et al. (2015) found an association between chronic diseases such as hypertension, arthritis, and diabetes and a poorer quality of life across several domains in BC survivors, leading to reduced physical and social functioning [41]. Furthermore, Pérez-Fortis et al. (2017) show that having other health issues greatly increases the unmet care needs of BC patients related to physical and daily living activities compared to people who do not have any other health issues [11]. Comorbidities can significantly impact the progression, diagnosis, treatment, and outcomes of breast cancer. These conditions may also exacerbate symptoms like fatigue, pain, and weakness, leading to increased physical challenges and impaired daily living activities.

The presence of comorbid conditions, which are prevalent in cancer patients, leads to worse symptoms, particularly pain [43]. Furthermore, even with painkillers, these patients continue to experience increased physical needs, which significantly impact their daily activities. Our study indicated that 59.9% of participants needed support in pain management. This finding is lower than local studies conducted in Gaza, where 75.5% [19] and 87.3% [25] of participants reported a need for support in pain management under the physical and daily living needs domain. In comparison to global studies, our findings reveal higher rates of pain. For example, research in Singapore showed that 36% of BC survivors suffered pain [26], while a study in Malaysia revealed that 55.6% of BC patients had significant pain-related needs [31]. Additionally, a systematic review using a large sample size (*n* = 1210) across 77 studies revealed a pain prevalence of 45.5% among BC patients [6]. Another systematic review of 19 studies highlighted fatigue and pain as the most common symptoms among BC survivors [44]. There is a shortage of painkiller medications in Palestinian cancer oncology hospitals, and the available ones are in limited supply. Patients and their families incur higher costs due to the inconsistent availability of these drugs, forcing them to purchase them independently [45]. This may indicate insufficient pain control, underscoring the need for health professional education and training to emphasize evidence-based pain management approaches and strategies, including regular pain assessment and reassessment [24,46].

### 4.4. Social Support and Supportive Care Domains

Our study, along with previous research, highlights the crucial role of family support for breast cancer patients, particularly in the Palestinian context. Family members, especially sons and daughters, were identified as the primary source of support for most participants. The relationship between social support and diverse areas of supportive care needs is well established [44]. Studies have consistently shown a strong association between social support and reduced unmet needs in various domains of supportive care. Schmid-Büchi et al. (2013) emphasized the importance of assessing the relationships between BC patients and their family members, highlighting the strong association between insufficient social support and unstable relationships, as well as the heightened need for supportive care [32]. In the United States, family, friends, husbands, and healthcare workers were identified as the main sources of support for breast cancer patients [47]. Similarly, a review of 77 studies found that social support was the most prevalent source of unmet needs, accounting for 74% of the total [6]. In Palestinian culture, family support plays a vital role in providing emotional, practical, and psychological support, helping patients navigate their illness, reduce isolation, and improve treatment adherence.

### 4.5. Study Implications

The findings of this study have important implications for the Palestinian healthcare system. The high unmet needs in the healthcare system and information domain underscore a lack of knowledge about breast cancer and its treatment among patients. This suggests a need for reforms in hospital organization and institutional policies to better address patient needs. Moreover, the significant prevalence of psychological needs highlights the importance of improving access to mental health services, especially for patients with anxiety and depression. Routine assessment of psychological distress and supportive care needs, alongside symptom assessment, is recommended to create a more supportive and understanding care environment. Additionally, healthcare providers should address both physical and psychological needs, provide clear information about the illness and treatment, and monitor mental distress. Empowering patients and fostering their engagement with the medical team can enhance their overall health and well-being. Finally, the Ministry of Health should integrate mental health services, patient education, and medical professional training into the healthcare system. This may require legislative and policy changes to ensure comprehensive care for breast cancer patients.

### 4.6. Study Limitations

This study has several limitations. First, conducting the study in a restricted geographical area of Palestine may limit the generalizability of the results. Second, the study’s cross-sectional approach restricts the capacity to make associations between the findings and unmet needs. Third, questions about sexuality may have resulted in informational bias because of cultural sensitivity towards this particular issue. Finally, including both new and old cases may introduce variability in patient experiences and illness stages, potentially affecting the consistency of findings and potentially causing underestimation or overestimation of unmet needs.

## 5. Conclusions

Our findings highlight a significant unmet need for supportive care services among breast cancer patients in Palestine. The most pressing needs were in the healthcare system and information, followed by physical and daily living, and psychological domains. These results suggest that existing healthcare services may not adequately address the full range of patient needs. Therefore, the Ministry of Health should prioritize the delivery of comprehensive supportive care, including specialized services from professionals trained in psychology, nutrition, and social work. However, our study revealed that these services were often unavailable in the settings where the research was conducted.

Moreover, the findings indicate that sociodemographic and clinical characteristics, as well as social support, significantly influenced unmet needs and supportive care domains. This underscores the importance of personalized care approaches. Further research, particularly longitudinal studies, is needed to understand the long-term effects of breast cancer on supportive care needs and the impact of information provided by healthcare professionals.

## Figures and Tables

**Figure 1 cancers-16-03663-f001:**
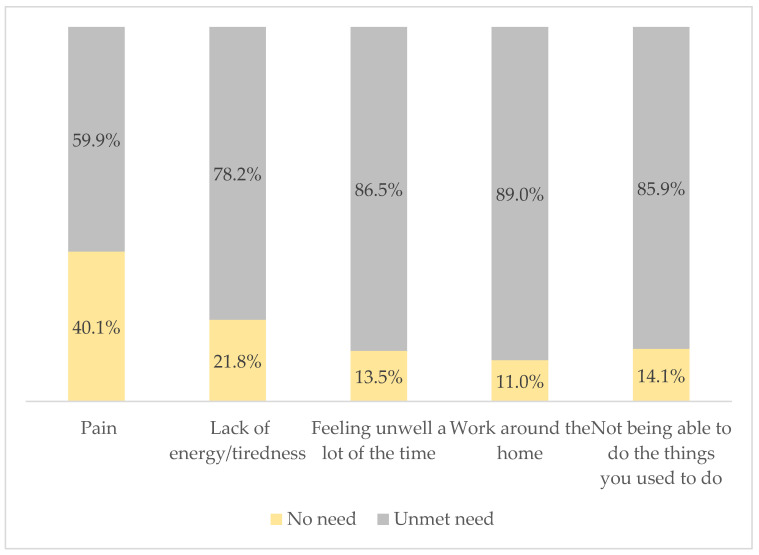
Physical and daily living needs (SCNS-34) of respondents.

**Figure 2 cancers-16-03663-f002:**
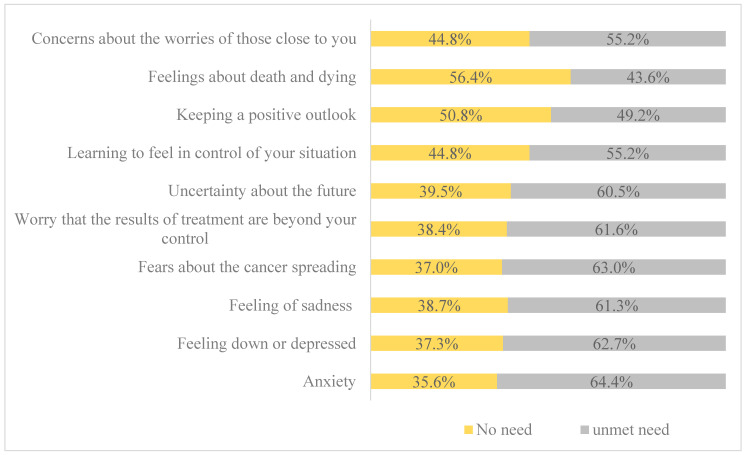
Psychological needs (SCNS-34) of respondents.

**Figure 3 cancers-16-03663-f003:**
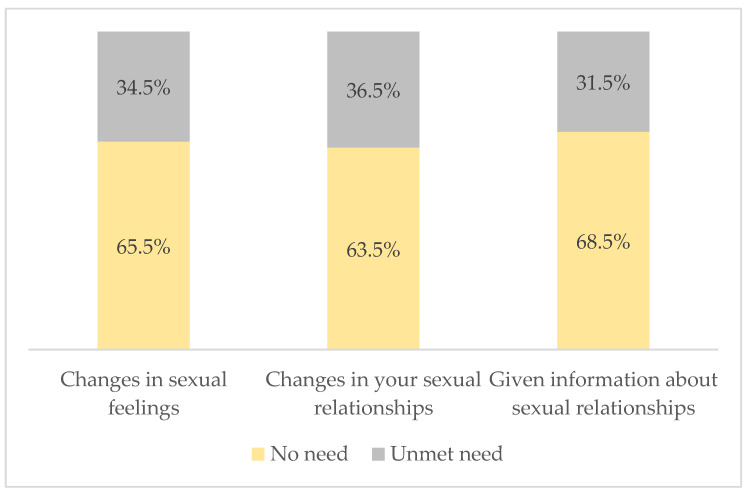
Sexual needs (SCNS-34) of respondents.

**Figure 4 cancers-16-03663-f004:**
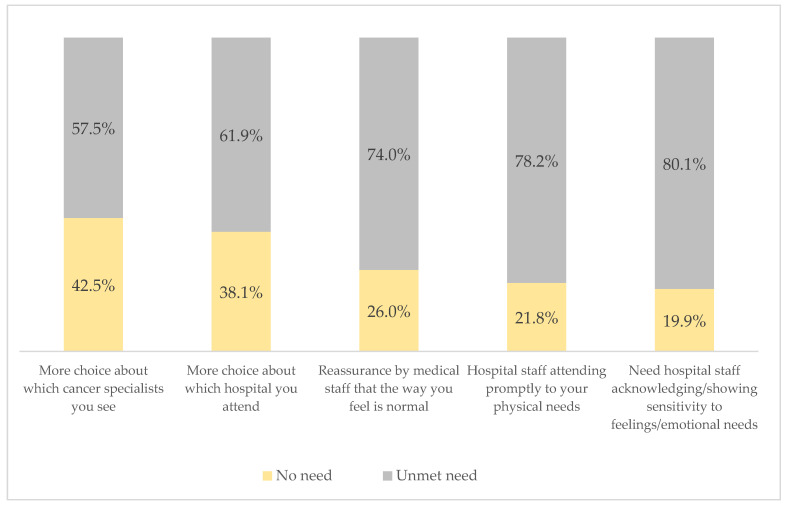
Patient support and care needs (SCNS-34) of respondents.

**Figure 5 cancers-16-03663-f005:**
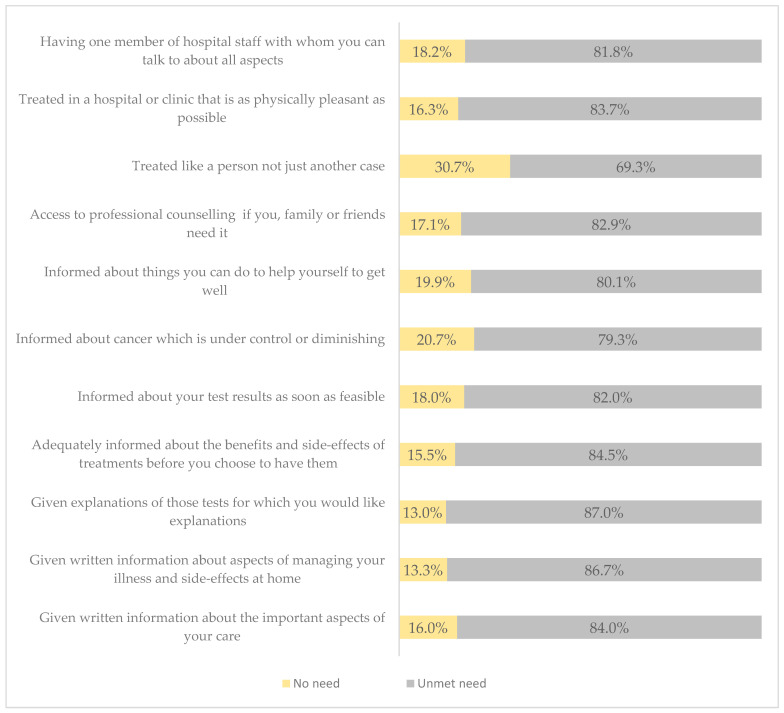
Health system and information needs (SCNS-34) of respondents.

**Figure 6 cancers-16-03663-f006:**
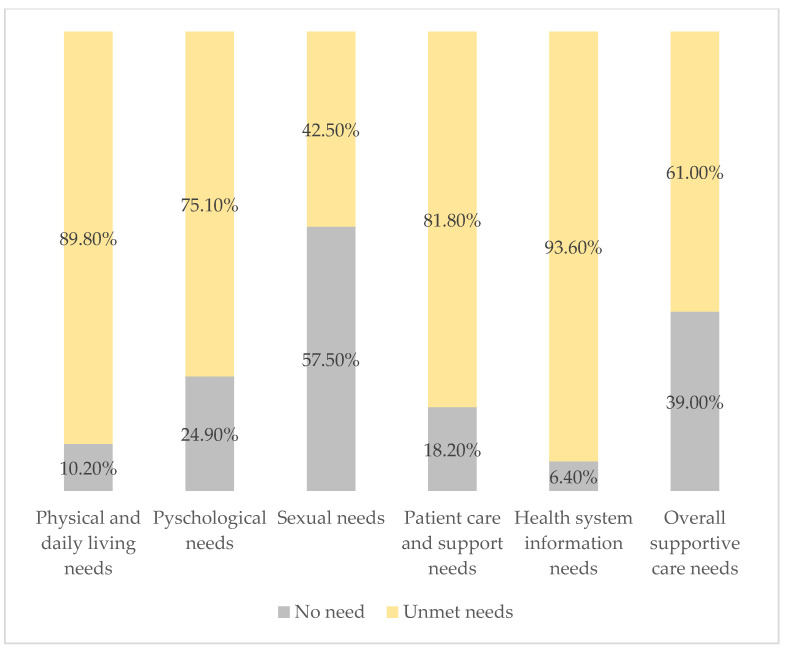
Prevalence of supportive care needs (SCNS-34).

**Table 1 cancers-16-03663-t001:** Sociodemographic and economic characteristics of participants (*N* = 362).

Variables	No.	%
**Age in years**		
≤40	78	21.5%
41–54	186	51.4%
≥55	98	27.1%
**Patient governorate**		
Hebron	244	67.4%
Bethlehem	118	32.6%
**Place of residence**		
Refugee camps	27	7.5%
Village	215	59.4%
City	120	33.1%
**Marital status**		
Unmarried	28	7.7%
Married	305	84.3%
Divorced/widow	29	8.0%
**Number of children**		
≤5 persons	245	67.7%
>5 persons	117	32.3%
**Family size (extended family)**		
≤5 persons	132	36.5%
>5 persons	230	63.5%
**Patient religion**		
Muslim	341	94.2%
Christian	21	5.8%
**Education**		
Primary and less	62	17.1%
Secondary	163	45.0%
Diploma	31	8.6%
University and above	106	29.3%
**Working status**		
Employee	73	20.2%
Housewife	280	77.3%
Retired	9	2.5%
**Monthly salary (USD)**		
<570	163	45.0%
570–1140	138	38.1%
>1140	61	16.9%

**Table 2 cancers-16-03663-t002:** Clinical characteristics and support for participants (*N* = 362).

Variables	No.	%
**Diagnosis duration (months)**		
Less than 6	90	24.9%
6–12	49	13.5%
More than 12	223	61.6%
**Stage of disease at diagnosis**		
Stage 1	24	6.6%
Stage 2	123	34.0%
Stage 3	172	47.5%
Stage 4	43	11.9%
**Type of treatment**		
Chemotherapy	347	95.9%
Radiotherapy	246	68.0%
Hormonal therapy	194	53.6%
Surgical therapy	295	81.5%
Biological treatment	136	37.5%
**Surgical intervention**		
Total mastectomy	119	40.2%
Partial mastectomy	177	48.9%
No surgical intervention	58	18.2%
**Chronic diseases other than cancer**		
No	236	65.2%
Yes	126	34.8%
**Taking pain medication**		
No	180	49.7%
Yes	182	50.3%
**History of relative cancer**		
Breast cancer	100	27.6%
Other cancers	82	22.7%
No cancer history	180	49.7%
**Referred to a non-cancer specialist**		
No	352	97.2%
Yes	10	2.8%
**Family and sons/daughters support**		
No	56	15.5%
Yes	306	84.5%
**Husband and partner support**		
No	269	74.3%
Yes	93	25.7%
**Medical staff support**		
No	356	98.3%
Yes	6	1.7%
**Other sources of support**		
No	279	77.1%+
Yes	83	22.9%

**Table 3 cancers-16-03663-t003:** Supportive Care Need (SCN-SF34) domains descriptive statistics.

Supportive Care Needs Items	No.	Mean	SD	Median	Min.	Max.	Range
Sexuality	362	6.31	3.36	6.00	3	15	12
Patient care support	362	16.34	5.16	16.00	5	25	20
Physical and daily living	362	17.98	5	19.00	5	25	20
Psychological	362	29.22	10.84	29.00	10	50	40
Health system information	362	39.64	10	41.00	11	55	44

SD: standard deviation, Min: minimum, Max: maximum.

**Table 4 cancers-16-03663-t004:** Multivariate analysis (logistic regression analysis).

Supportive Care Needs Domains		95% CI		
	AOR	Lower	Upper	*p*-Value
**Physical and daily living care needs**				
Age (years)				
≤40	0.21	0.06	0.73	0.01
41–54	0.31	0.09	0.99	0.04
≥55	**Reference**			
Do you take pain medication? (Yes/No)	5.36	2.19	13.06	<0.001
Chemotherapy therapy (Yes/No)	4.49	1.08	18.55	0.03
Surgical therapy (Yes/No)	5.15	1.52	17.42	0.00
Family and sons/daughters support (Yes/No)	0.19	0.04	0.89	0.03
**Psychological care needs**				
Age (years)				
≤40	13.37	5.23	34.16	<0.001
41–54	5.84	3.29	10.38	<0.001
≥55	**Reference**			
Hormonal therapy (Yes/No)	0.361	0.20	0.63	<0.001
**Sexual care needs**				
Age (years)				
≤40	36.90	15.05	90.42	<0.001
41–54	5.40	2.69	10.82	<0.001
≥55	**Reference**			
Marital status				
Single	0.87	0.18	4.08	0.8
Married	4.62	1.44	14.82	0.01
Divorced/widow	**Reference**			
Family relative history of cancer				
Breast cancer	2.34	1.27	4.30	0.006
Other cancers	1.23	0.64	2.37	0.52
No cancer history	**Reference**			
Radiotherapy therapy (Yes/No)	0.42	0.24	0.74	0.003
Biological therapy (Yes/No)	0.40	0.22	0.70	0.001
Governorate				
Hebron	0.52	0.30	0.90	0.02
Bethlehem	**Reference**			
**Patient care and support care needs**				
Age (years)				
≤40	4.54	1.85	11.18	0.001
41–54	3.18	1.73	5.84	<0.001
≥55	**Reference**			
Monthly Income (USD)				
<570	0.35	0.13	0.90	0.03
570–1140	0.62	0.23	1.68	0.35
>1140	**Reference**			
Hormonal therapy (Yes/No)	0.48	0.26	0.881	0.01
**Health information care needs**				
Age (years)				
≤40	5.30	1.15	24.44	0.03
41–54	2.74	1.11	6.76	0.02
≥55	**Reference**			
**Total care needs**				
Age (years)				
≤40	8.20	3.81	17.61	<0.001
41–54	2.34	1.37	4.00	0.002
≥55	**Reference**			
Do you take pain medication? (Yes/No)	2.01	1.24	3.26	0.004
Hormonal therapy (Yes/No)	0.45	0.27	0.75	0.002
Biological therapy (Yes/No)	0.47	0.28	0.80	0.006
Surgical intervention				
Complete mastectomy	1.31	0.65	2.63	0.44
Partial mastectomy	2.27	1.18	4.35	0.01
No surgical intervention	**Reference**			
Family and sons/daughters support (Yes/No)	0.50	0.25	1.00	0.05

## Data Availability

The original contributions presented in the study are included in the article/Appendix A; further inquiries can be directed to the corresponding author.

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
