# Peer review of "Factors Associated with Supportive Care Needs Among Palestinian Women with Breast Cancer in the West Bank: A Cross-Sectional Study"

_cancers, 2024, doi:10.3390/cancers16213663_

Round 1
Reviewer 1 Report
Comments and Suggestions for Authors
This is a welcome manuscript on supportive care needs among patients with breast cancer in Palestine. The authors use an appropriate instrument to measure supportive care needs (SCNS-SF34) and effectively used survey methodology to address their research questions.
Some methodological points:
* When were patients recruited? Over what span of time? Understanding the calendar time when patients were recruited is helpful to understand clinical context.
* I see several references to biological therapy, but am unclear what falls under that category. For example, does biological therapy include immunotherapy? Please provide a bit more detail on that category.
* If possible, it would be helpful to present median values alongside mean/sd/min/max/range. However, this is not critical.
* Power analysis assumed a potential non-response rate of 10%. What was the actual non-response rate?
* Did all participants complete all survey items? Please confirm what missingness was observed across survey elements, possibly in the supplemental material if needed.
* The Figure 2 y axis has a label "Feeling of sadness feelings", I think this should simply read "Feelings of sadness".
* Why do the authors consistently refer to "families and sons" instead of simply "families"? I would assume sons are part of families, and now wonder whether daughters are also included as part of "families" or if they are excluded for some reason. Please clarify.
Discussion:
The authors write "The findings of this study may reveal a substantial deficiency in the existing supportive
care services at the Palestinian Ministry of Health (MoH) facilities.". To what extent is there capability for improvement by the MoH? What barriers might the MoH face in the implementation of the care that is suggested in subsequent sentences?
Conclusions:
The authors write "The study emphasizes the importance of customizing supportive care services to meet the specific needs of breast cancer patients in Palestine". For clarity in this section, please provide an example or broad explanation of _what_ supportive care services would address _which_ specific needs.
Author Response
Dear Reviewer
Thank you for reviewing our manuscript. The comments are very important. To respond to all comments, the responses for each are in red below. We also added the required changes and additions in red in the manuscript.
Reviewer 1
This is a welcome manuscript on supportive care needs among patients with breast cancer in Palestine. The authors use an appropriate instrument to measure supportive care needs (SCNS-SF34) and effectively used survey methodology to address their research questions.
Some methodological points:
* When were patients recruited? Over what span of time? Understanding the calendar time when patients were recruited is helpful to understand clinical context.
Thank you for your comment. We added the timing of the survey Line 124
“Data was collected between March and July 2024”.
* I see several references to biological therapy, but am unclear what falls under that category. For example, does biological therapy include immunotherapy? Please provide a bit more detail on that category.
Please see line 316:
Yes, we mean by biological therapy the immunotherapy therapy.
* If possible, it would be helpful to present median values alongside mean/sd/min/max/range. However, this is not critical.
Thank you for your comment. Median values are added to the table.
* Power analysis assumed a potential non-response rate of 10%. What was the actual non-response rate?
We were lucky that only 2 women refused to participate. Please read the sentence line 118.
* Did all participants complete all survey items? Please confirm what missingness was observed across survey elements, possibly in the supplemental material if needed.
No, we did not have lost items, the survey was an interview and the researcher spent a lot of time with the participants and she participated in all activities they had which built trust with them.
* The Figure 2 y axis has a label "Feeling of sadness feelings", I think this should simply read "Feelings of sadness".
Thank you for your comment. We corrected it
* Why do the authors consistently refer to "families and sons" instead of simply "families"? I would assume sons are part of families, and now wonder whether daughters are also included as part of "families" or if they are excluded for some reason. Please clarify.
Thank you for your note. Yes, it is her sons and daughters. It is a translation mistake of the Arabic term. We modified the term to sons/daughters in the whole manuscript.
Discussion:
The authors write "The findings of this study may reveal a substantial deficiency in the existing supportive care services at the Palestinian Ministry of Health (MoH) facilities.". To what extent is there capability for improvement by the MoH? What barriers might the MoH face in the implementation of the care that is suggested in subsequent sentences?
Thank you for your comment.
Please see Line 383. We added the following:
Although the MoH is responsible for providing cancer care, the political conflict in the region has resulted in a deterioration in the Ministry of Health's financing, putting the provision of essential health services for cancer patients at risk due to economic constraints. Therefore, collaboration with other healthcare providers or institutions can enhance the delivery of comprehensive services in regions where the Ministry of Health faces service shortages. Additionally, receiving support from donor organizations through financial and technical assistance will be advantageous
Conclusions:
The authors write "The study emphasizes the importance of customizing supportive care services to meet the specific needs of breast cancer patients in Palestine". For clarity in this section, please provide an example or broad explanation of _what_ supportive care services would address _which_ specific needs.
Please see Line 627, we added the following:
Therefore, the Ministry of Health should prioritize the delivery of comprehensive supportive care, including specialized services from professionals trained in psychology, nutrition, and social work. However, our study revealed that these services were often unavailable in the settings where the research was conducted.

Reviewer 2 Report
Comments and Suggestions for Authors
This manuscript evaluates the effectiveness of the supportive care service provided to patients with breast cancer (BC) living the West Bank of Palestine. To this aim, a well-designed questionnaire was constructed to data collection (on sociogeographic, clinical and social support factors) and participants have been interviewed by the medical staff. Results revealed that a significant proportion of the participants was not completely satisfied by the supportive care services and that these services need to be customized to meet the specific needs of BC patients in Palestine.
My comments are reported below.
Introduction: It would be better to indicate breast cancer with the most conventional abbreviation (BC) instead of (breast ca).
Please, check reference n.2 and update it with “Global cancer statistics 2022: GLOBOCAN estimates of incidence and mortality worldwide for 36 cancers in 185 countries”.CA Cancer J Clin. 2024;74:229–263.
Materials and Methods:
2.1
line 90 use BC instead of breast cancer.
Line 94 there is a typing error
2.4
I think that it should be useful to add a sample sheet of the questionnaire
Line 140 Please, remove the red dot
Results:
Line 187 Please, uniform Table 1 caption with the description in the text
Table 3. line 215. I would suggest to indicate supportive care needs items starting from the most significative one to the less significative (from Health system information to sexuality)
Discussion:
Discussion needs to go more in detail into certain areas. Using many different paragraphs without clear connections makes the text not fluent. Authors should better argue this section.
Comments on the Quality of English Language
The authors are encouraged to get editing help from someone with full professional proficiency in English.
Author Response
Dear Reviewer
Thank you for reviewing our manuscript. The comments are very important. To respond to all comments, below you will find the responses for each in red. We also added the required changes and additions in red in the manuscript.
Reviewer 2
This manuscript evaluates the effectiveness of the supportive care service provided to patients with breast cancer (BC) living the West Bank of Palestine. To this aim, a well-designed questionnaire was constructed to data collection (on sociogeographic, clinical and social support factors) and participants have been interviewed by the medical staff. Results revealed that a significant proportion of the participants was not completely satisfied by the supportive care services and that these services need to be customized to meet the specific needs of BC patients in Palestine.
My comments are reported below.
Introduction: It would be better to indicate breast cancer with the most conventional abbreviation (BC) instead of (breast ca).
Thank you for the comment. We used BC now instead of breast ca. Changes are highlighted with a red color
Please, check reference n.2 and update it with “Global cancer statistics 2022: GLOBOCAN estimates of incidence and mortality worldwide for 36 cancers in 185 countries”.CA Cancer J Clin. 2024;74:229–263.
Materials and Methods:
2.1 line 90 use BC instead of breast cancer.
Done
Line 94 there is a typing error
Error deleted
2.4 I think that it should be useful to add a sample sheet of the questionnaire
Line 140 Please, remove the red dot
Dot removed
Results:
Line 187 Please, uniform Table 1 caption with the description in the text
Table 3. line 215. I would suggest to indicate supportive care needs items starting from the most significative one to the less significative (from Health system information to sexuality)
Thank you for this suggestion. In accordance with your suggestion, we modified the table and added the median values.
Discussion:
Discussion needs to go more in detail into certain areas. Using many different paragraphs without clear connections makes the text not fluent. Authors should better argue this section.
Thank you for your comment
We modified the discussion part and added some more justification to the different parts. We could not do it with track changes since we did many modifications. Some of the major modifications are in red
Comments on the Quality of English Language
The authors are encouraged to get editing help from someone with full professional proficiency in English.
We did full document editing by an English language editor
Round 2
Reviewer 2 Report
Comments and Suggestions for Authors
Introduction, Lines 44-45 The suggested changes have not been made. Reference 2 is still the old one. See line 657
Author Response
Dear Reviewer
Thank you for reviewing again our manuscript. Sorry, we missed replacing the reference
Please, check reference n.2 and update it with “Global cancer statistics 2022: GLOBOCAN estimates of incidence and mortality worldwide for 36 cancers in 185 countries”.CA Cancer J Clin. 2024;74:229–263.
- The new reference is added in red line 655.
“Bray F, Laversanne M, Sung H, Ferlay J, Siegel RL, Soerjomataram I, Jemal A. Global cancer statistics 2022: GLOBOCAN estimates of incidence and mortality worldwide for 36 cancers in 185 countries. CA Cancer J Clin. 2024 May-Jun;74(3):229-263. doi: 10.3322/caac.21834”
- Also, we updated the data lines 44-45
“In 2022, there were an estimated 2.3 million newly diagnosed cases with BC and 660,000 related deaths globally [2]. “